# Fault Injection with Multiple Fault Patterns for Experimental Evaluation of Demand-Controlled Ventilation and Heating Systems

**DOI:** 10.3390/s22218180

**Published:** 2022-10-25

**Authors:** Bahareh Kiamanesh, Ali Behravan, Roman Obermaisser

**Affiliations:** Department of Electrical Engineering and Computer Science, University of Siegen, 57076 Siegen, Germany

**Keywords:** multiple faults, fault injection framework, fault model, fault occurrence probability, scenario generation, HVAC system, MATLAB

## Abstract

Heating, ventilation, and air-conditioning (HVAC) systems are large-scale distributed systems that can be subject to multiple faults affecting the electronics, sensors, and actuators, potentially causing high energy consumption, occupant discomfort, degraded indoor air quality and risk to critical infrastructure. Fault injection (FI) is an effective experimental method for the validation and dependability evaluation of such HVAC systems. Today’s FI frameworks for HVAC systems are still based on a single fault hypothesis and do not provide insights into dependability in the case of multiple faults. Therefore, this paper presents modeling patterns of numerous faults in HVAC systems based on data from field failure rates and maintenance records. The extended FI framework supports the injection of multiple faults with exact control of the timing, locality, and values in fault-injection vectors. A multi-dimensional fault model is defined, including the probability of the occurrence of different sensor and actuator faults. Comprehensive experimental results provide insights into the system’s behavior for concrete example scenarios using patterns of multiple faults. The experimental results serve as a quantitative evaluation of key performance indicators (KPI) such as energy efficiency, air quality, and thermal comfort. For example, combining a CO_2_ sensor fault with a heater actuator fault increased energy consumption by more than 70%.

## 1. Introduction

Buildings are responsible for 40% of global energy usage and contribute 30% of the total CO_2_ emissions [1]. Typically, 20–30% of energy savings in buildings are achievable by recommissioning the HVAC systems to rectify faulty operations [2,3]. HVAC systems are the main reason for global energy consumption, electricity consumption, and worldwide CO_2_ emissions [4]. For example, the construction and maintenance of building stock are responsible for 36% of the CO_2_ emission in the European Union (EU) [5]. In 2018, the building, construction, and building processes represented 36% of energy consumption, 39% of energy-related CO_2_ emissions, and 50% of global electricity consumption [4,6]. In critical infrastructure such as airports and hospitals, heating, ventilation, and air-conditioning (HVAC) systems also play a prominent role in emergency scenarios (e.g., fires, biological hazards).

Faults in HVAC systems can cause temperature fluctuations, occupancy discomfort, excess ventilation, and overheating. Fault management is a significant component of a building management system (BMS) for mitigating faults and their high-level symptoms [7,8]. For example, Teraoka et al. [7] proposed a fault management framework, BDSherlock, based on two lists. One list comprises standard checks, and the other contains rules based on anomalies. They use data-driven analysis techniques to investigate the energy impact of the detected faults on the HVAC system. 

Fault injection (FI) is an effective experimental method for the validation and dependability evaluation of HVAC systems with such fault management techniques. FI provides insights into the system’s behavior by deliberately introducing faults in different scenarios and conditions. FI frameworks for HVAC systems have been introduced in previous work [8], and the impact of individual faults on system behavior has been evaluated. However, many HVAC systems (e.g., hospitals, airports, multi-story office buildings) are large-scale distributed systems with thousands of components, including sensors, actuators, computational nodes, and communication links which are vulnerable and prone to faults. Therefore, FI must investigate the effects of multiple faults simultaneously. This is in significant contrast to smaller scale systems (e.g., automotive electronics, medical equipment) where a single fault hypothesis is predominant [9] and considering a single fault at a given point in time is sufficient. Today’s FI frameworks for HVAC systems are based on this single-fault hypothesis. However, systems face multiple faults, in reality [10].

Therefore, this paper provides contributions toward a dependability evaluation of HVAC systems faced with multiple faults at the same time: 

**Modeling patterns of multiple faults in DCV and heating systems based on data from field failure rates and maintenance records:** The paper maps insights from maintenance records to FI patterns with multiple faults. The fault occurrence probability is an important parameter in the design of a realistic FI framework because this parameter is affected by environmental conditions, e.g., dust and dirt, seasons and respective temperatures, working conditions, application areas, and the locality of faults in various components of a system. Therefore, the fault model is created using statistical parameters such as fault occurrence probability. Fault occurrence probabilities enable the definition of scenarios and the performance of FI based on different environmental conditions and fault type rates.

**Injecting multiple faults into a DCV and heating system according to the fault models:** The paper introduces an FI framework where faults are activated by an FI vector that precisely controls the attributes of multiple faults such as timing, locality, type, persistence, and values. The designed FI framework injects multiple faults into multiple zones and multiple components with corresponding fault attributes. An automatic FI algorithm initiates the fault attributes. Fault repetitions and multi-dimensional fault attributes are assigned in a randomized manner. The framework is generic, and the matrices can be customized for different structures and buildings. The paper shows how fault patterns for multiple faults can be established for a particular structure and environmental conditions based on maintenance records from prior work. 

**Experimentally evaluating the effects of multiple faults on the behavior of the DCV and heating systems:** The paper provides comprehensive experimental results and insights into the system behavior upon multiple faults using patterns of multiple faults. Due to the use of real-world data and maintenance records, the results are realistic. This is a significant result of research on fault management techniques coping with multiple faults, for which no experimental data is available today.

## 2. Related Work 

### 2.1. Fault Injection in HVAC Systems

For the assessment of quality constraints such as resource usage, resource availability [10], thermal conditions, occupant comfort, and dependability of a system under faults, different approaches, including analytical modeling [11] and experimental methods such as FI [12,13] are discussed in the literature. FI brings high controllability and observability in a simulation environment. FI techniques are classified into four categories: (1) physical FI, including hardware-based FI and software-based FI methods; (2) simulation-based FI methods; (3) emulation-based FI methods; and (4) hybrid FI methods [14,15]. Arlat et al. [16] have introduced an FI methodology for two main goals: validation and design aid. They have also described different modeling abstraction levels, including axiomatic, empirical, and physical models. Axiomatic models emphasize analytic models such as Markov graphs and fault trees. Empirical models relate to more complex and detailed behavior and structural descriptions, such as simulation and physical models implemented as hardware and software features. Kiamanesh et al. [8] discussed FI methods and presented a realistic, simulation-based FI framework for evaluating DCV and heating systems. Their introduced FI combines two techniques, simulator command, and simulation code modification, implemented with Stateflow for single faults. 

### 2.2. Experimental Evaluation in HVAC Systems

The experimental evaluation of HVAC systems in the design phase is an important subject [17] to enhance the system’s efficiency, resource usage [18], economic effectiveness [19], thermal comfort [20], and reduce CO_2_ emissions [6,18,19,20,21,22]. Extensive research has presented experimental evaluations of energy consumption for HVAC systems. Antonopoulos et al. [23] proposed an experimental assessment of the energy savings of air conditioning (A.C.). Al-Deen et al. [24] evaluated the energy consumption of HVAC systems under different climate conditions. Vishwanath et al. [4] investigated the HVAC cooling energy consumption and cost associated with experiments conducted in large buildings. Andrés et al. [20] performed a real-scale experimental evaluation for regulating thermal control in lightweight constructions. Krajcik et al. [25] performed an experimental evaluation of residential rooms for sustainable heating/cooling and efficient energy consumption. Arteconi et al. [26] introduced an experimental assessment of a ground coupled heat pump (GCH), an alternative to traditional systems for heating and cooling. 

### 2.3. Multiple Faults in HVAC Systems and Other Domains

Multiple faults have been investigated in domains other than HVAC systems. Yalcin et al. [27] have injected different hardware faults, such as transient, intermittent, permanent, and multi-bit faults, in simulations of processors. Multi-bit faults occur when a fault affects multiple bits simultaneously, such as spatial multi-bit upsets. Stroud et al. [28] have described single and multiple stuck-at-fault simulations for gate-level faults. Multiple faults are injected randomly or clustered for testing multiple fault detection capabilities. A list of fault groups has been considered for injecting multiple faults. Each fault group contains a number of gate-level stuck-at faults with a number of potential fault sites and possible combinations of single and multiple stuck signals at the gate level. Faults are injected randomly or in a cluster-based manner. The selection can be changed from a random sample to a deterministic function in the clustered FI. It modifies for clustered defects that tend to form a list of faults that are tightly coupled based on the degree of the cluster. Tarrilo et al. [29] introduced a multiple-bit-flip FI platform. They triggered multiple faults in SRAM-based FPGAs, which are sensitive to soft errors, unexpected bit-flips, and critical errors. They injected single-event upsets (SEUs) and multiple-bit upsets (MBUs) for functional errors. The location of each malfunction chooses from a list of locations. Kundu et al. [30] injected multiple faults to diagnose chips at the logic level. Arlat et al. [31] compared physical and software-based FI for the MARS fault-tolerant distributed real-time system. They addressed the respective impacts of FI techniques using a testbed and test scenarios. Zhong et al. [32] investigated operational single and multiple-fault impacts for HVAC systems under different climates. The effect of faults in HVAC systems may depend on climate changes. They also evaluated the system’s impacts on thermal comfort, performance, and energy usage. They ranked single and multiple faults for each climate condition. However, they did not carry out simulation-based multiple FI. Sangchoolie et al. [33] evaluated the impacts of single and multiple bit-flip errors. They used the open-source fault injector tool LLFI, which injects faults into the low-level virtual machine (LLVM). To realize the injection of multiple faults, they extended LLFI to facilitate the injection of multiple bit-flips. LLFI defines single bit-flip errors as time location pairs. To model multiple bit-flip errors, they developed the time location parameters that enable clustering of the error space into different classes of errors. Tadeusiewicz et al. [34] introduced a method for simulating multiple faults in AC circuits. They used a systematic approach to perform the combination of multiple faults. The FI procedure uses admittance and impedance matrices for the faulty circuit nodes and fault combinations. Lisboa et al. [35] described soft errors that may appear at the same time. Robust operators are introduced, and the operator’s behavior is analyzed by simulating single and multiple faults. Papadimitriou et al. [36] introduced a multiple-fault injection methodology for digital circuits. Fault modeling at the register transfer level (RTL) can occur early in the design phase and facilitates the analysis of the gate-level models. They injected multiple faults by partitioning the RTL description of the circuits. Then, faults inject into two categories. Firstly, faults inject into one or more flip-flops, and the second group includes faults occurring in the combinational part of the circuits. Wang et al. [37] discussed hierarchical model-based diagnosis (MBD) for multiphase faults and hitting calculation sets (MHS), which serve for stability and reliability in power distribution networks. They calculated the system performance when the distributed network has multiple multiphase faults. The hierarchical MBD comprises different parts including an offline model library, fault observations, and online identification of faulty elements. Takahashi et al. [38] introduced and simulated the diagnosis of single and multiple faults in combinational circuits. Kim et al. [39] introduced the modeling and simulation of multiple faults. The multiple fault model consists of a set of lines. For example, the stuck-at fault consists of two lines stuck-at-1 and stuck-at-0. Any fault combination can be modeled by activating these lines.

### 2.4. Fault Occurrence Probabilities in HVAC Systems

Dynamic variables such as outdoor weather conditions and indoor occupant behaviors influence HVAC systems. Each has its associated occurrence probability. Stochastic approaches result in a better and more realistic simulation by showing different occurrence probability values in various components under diverse conditions. However, defining a fault occurrence probability distribution and a mathematical expression for other faults is challenging for HVAC systems due to the different types of faults, component fault rates, and environmental conditions. Some studies have suggested fault occurrence probabilities based on available maintenance and statistical records [40]. For example, Li et al. [41] reviewed fault modeling of HVAC systems in buildings and discussed fault occurrence probability distributions in HVAC systems. They provided probability tables for different fault types that describe each component’s fault occurrence rate. Myrefelt et al. [42] introduced stochastic equations at the component and system level using a sizeable operational dataset for HVAC components. Otto et al. [43] proposed an approach based on the probability density function (PDF), such as normal, skewed, quantile-based, and Weibull distributions [1]. This paper provides the occurrence probabilities based on the values suggested in these mentioned papers.

### 2.5. Research Gap and Contributions Discussion

An experimental evaluation for the simulation-based multiple FI framework in DCV and heating systems has been carried out in this paper, considering a comprehensive fault model where faults are activated with realistic fault patterns and combinations for different environmental conditions. The related work section explains various research fields considered in this paper. The contributions are discussed in the following with respect to the research gaps.

#### 2.5.1. Modeling Patterns of Multiple Faults in DCV and Heating Systems Based on Data from Field Failure Rates and Maintenance Records

In prior research [40,41,42,44,45], fault detection and diagnosis (FDD) techniques were introduced with fault attributes derived from maintenance records of HVAC systems. However, only individual faults were addressed whereas the consideration of combinations of faults is essential for large-scale electronic systems. This paper provides contributions beyond the state-of-the-art by introducing fault models and patterns for combinations of multiple faults, which consider fault attributes (e.g., occurrence rates, locality, persistence) from maintenance records and serve for FI and FDD in HVAC systems. Each fault combination has a specific occurrence rate based on the fault attributes, such as fault types. The fault model and occurrence rate are compatible with different environmental conditions, by mapping the fault occurrence to the real-world maintenance records.

#### 2.5.2. Injecting Multiple Faults into a DCV and Heating System 

In previous works, individual faults were injected into HVAC systems [8,12,13,46,47] and the injection of multiple faults was considered in other domains, such as semiconductor technology [10,27,30,31,34,35,37,39]. Hence, the injection of multiple faults with corresponding attributes is a research gap for DCV and heating systems. This paper goes beyond the state of the art by introducing a framework for injecting multiple faults with corresponding fault attributes, while observing the propagation of the faults from the component level to the system level and the manifestation of system-level properties (e.g., energy efficiency, occupant comfort). The introduced framework is generic and scalable, and it can be instantiated for different building structures and fault combinations. The fault attributes are expressed using matrices, which are extended in their size and their dimensions to support more complex structures with additional components, zones and buildings. The FI occurs using an HVAC simulation framework with realistic physical models of thermodynamics, heat/air flow transfer and environmental conditions.

#### 2.5.3. Experimentally Evaluating the Effects of Multiple Faults on the Behavior of DCV and Heating Systems

Experimental evaluations of HVAC systems were carried out in [19,20,23,24] to monitor the system behavior in the presence of faults. However, in the field of DCV and heating systems, the experimental evaluations of multiple faults in combination with different environmental conditions have not been published and no such experimental data is available. This research gap is a barrier for the development of fault-tolerance techniques and the dependability evaluation of HVAC systems. The FI framework introduced in this paper monitors the system behavior for different fault patterns and multiple fault combinations that are defined by the user. The FI framework is generic and enables the evaluation of quality attributes such as heating cost, energy consumption, occupant comfort, indoor temperature and air quality. 

## 3. System Model

This section introduces the system model of a DCV and heating system, including its components, fault classifications, and fault propagation. In addition, the FI framework and its features, including the fault profiles with the considered fault attributes, are described. 

### 3.1. HVAC Systems

DCV realizes a control strategy based on ventilation to moderate the amount of fresh air. It also optimizes the air quality in terms of CO_2_ concentration and temperature and balanced energy consumption by an automatic adjustment of the volume of the air exchange. It uses damper actuators according to the captured sensor measurements and values from air quality sensors and the environment [48].

Figure 1 illustrates an example building plan consisting of several floors. In addition, a part of the floor shows six typical rooms and a corridor equipped with a DCV and heating system [48]. Each room is generally equipped with several components, e.g., temperature sensors, CO_2_ sensors, occupancy sensors, damper actuators, and heater actuators.

Table 1 lists significant parameters such as the components’ temperature and CO_2_ signal thresholds and operational statuses. These parameters introduce the component conditions. They are later used for evaluating the system and scenario definitions.

### 3.2. Fault Classifications in HVAC Systems 

Inaccurate measurements due to hardware faults are inevitable in HVAC systems and lead to more energy consumption and low air quality. Bondavalli et al. [49] classified physical faults into two categories (1) permanent and (2) temporary faults. Permanent faults lead to abnormal behavior and wrong signals which continue constantly. The respective component should be removed or repaired to handle a permanent fault. Temporary physical faults are classified into internal (usually intermittent) and external (transient). An intermittent fault occurs regularly and continuously at the exact location, while a transient fault arises at random locations [49]. Many reasons exist for intermittent faults in different systems. Wakil et al. [50] discussed various intermittent fault causes in embedded electronic modules and explained that most of them are caused by interconnections and marginal design, e.g., loose or corroded wires, cracked solder joints, corroded or loose connectors, and broken wires [50,51]. Layali et al. [52] mentioned that the primary cause of intermittent faults is device wear out or the tendency of solid-state to degrade with time, stress, and time-dependent dielectric breakdown (TDDB), supposing the stress conditions persist in the long term.

Such faults may eventually lead to permanent defects. Different transient and intermittent faults include short transients, long transients, and short intermittent faults. Intermittent faults may disappear or become permanent [47]. Faults in HVAC systems can also be distinguished based on the design, developmental and operational phases in HVAC systems. For example, Torabi et al. [53] reviewed common human-made errors in different stages of creation in HVAC systems with multiple zones: preconstruction, construction, and operation phases. Frank et al. [54] discussed common faults and their relevance in the design and operation stages for HVAC systems, rooftop units (RTU), lighting, and refrigeration faults. The phase of a fault denotes when a fault occurs, e.g., during the design, development, or operational time of a system’s life cycle:**Developmental Faults**: A developmental fault occurs before the equipment installation. Developmental faults can be physical faults in production (e.g., inaccurate mask alignment) or design faults (e.g., incorrect positioning of sensors, improper scheduling of operations).**Operational Faults**: An operational fault occurs after the equipment installation phase. An example is wear out electronic components.

Developmental faults in a component can result in multiple faults in that component. It can also be considered a fault for other components commonly denoted as fault containment regions (FCRs) of the system. It shows that a root fault in a component leads to failure of that component which is a fault for other components or the whole system, which can lead to a system failure. Table 2 illustrates examples of fault propagation in HVAC systems. This paper considers each fault an event that occurs based on the probabilities of multiple faults. Faults in FCRs, such as inappropriate programming and improper setpoints, lead to component faults and potentially propagate to the system level. Each row of Table 2 presents the fault propagation from the component level to system level faults and the failure impact, such as energy waste or occupant discomfort.

### 3.3. Fault Injection Framework for Multiple Component Faults

An appropriate fault model that covers the relevant fault attributes is required to implement the FI framework. The fault model comprises the fault type, persistence, duration, interarrival time, fault location, fault occurrence rate, and fault repetitions described below.

#### 3.3.1. Fault Model

HVAC systems are integrated from numerous components, and due to their complexity, many kinds of faults and errors can emerge. This paper focuses on the faults of sensor and actuator components. To investigate these component faults (representing system faults) and their consequences on the system behavior, an FI framework with a complete fault profile is required, which fits the system model and introduces the attributes of each fault. A fault profile with its attributes, including fault type, fault persistence, fault duration time, fault interarrival time, fault repetition, and fault location, was described in [13]. This paper considers a new attribute named fault occurrence rate, parameter number 6 of Table 3. In all fault types, faulty values are produced by Equation (1) with defined relative coefficients.
(1)$$x^{\prime }=\beta x+\alpha +\eta$$
where,

*x* represents healthy data.*x*’ calculates faulty data.*β* is the coefficient of gain faults.*α* is the coefficient of offset faults. 

**Table 3 sensors-22-08180-t003:** Fault profile consisting of all required attributes for the FI framework.

Nr.	Fault Attributes	Descriptions
1	Fault type	Offset, gain, stuck-at, out of bounds, data loss, and white noise
2	Fault persistence	Short intermittent, and permanent
3	Fault duration	Fault presence period
4	Fault interarrival time	The time interval between faults
5	Fault location	Temperature sensor, CO_2_ concentration sensor, damper actuator, heater actuator
6	Fault occurrence rate	Probability of occurrence in case of each fault type
7	System fault repetition	Number of repetitions in case of each intermittent FI

##### Fault Types in HVAC Systems

Different fault types have been considered in our FI framework, stuck-at-faults for actuators and all six mentioned faults for sensors.

**Offset fault**: A shift value that adds to the actual sensed data due to sensing units’ bias/drift/calibration error. This fault has only been considered for the sensors.**Gain fault**: A gain fault occurs once the change rate of sensed data is different from the expected rate. This fault has only been considered for the sensors.**Stuck-at fault**: Stuck-at faults may happen in both actuators and sensors as stuck-at sensed values, stuck-at closed status in damper actuators, and stuck-at-off in heater actuators.**Out-of-bounds fault**: There are maximum and minimum bounds for each sensor, and sensor measurements should be in these ranges. There is an out-of-bounds fault when the observed values are out of the bounds of the expected values.**White noise fault**: A random number is added to the value, which is determined by Gaussian or uniform probability distributions [8].**Data loss fault**: There is missing data during a specific time interval. In case of a data loss fault, the last measurement of the sensed data indicates that the actual measurement is missed.

##### Fault Persistence in HVAC Systems

Different types of fault persistence have been considered in our proposed FI framework, permanent faults and short intermittent faults for actuators and only permanent faults for the sensors. A complete description of the fault attributes is explained below.

**Short and long intermittent faults**: Permanent and transient faults differ from intermittent faults, and few fault models describe intermittent faults with their occurrence frequencies [47,49,50,51,52,57]. Therefore, there is no reliable timing model for intermittent faults for the sensors in HVAC systems. Intermittent faults are more common in actuators. The timing parameters in the literature have been applied in our fault model [8,58,59]. Short intermittent faults have limited the fault repetitions to two repetitions. In the case of long intermittent faults, repetitions can increase based on the designer’s necessities, but faults also disappear eventually. In this paper, intermittent faults have been considered only for the actuators due to the lack of proper timing models for the sensors in HVAC systems.**Permanent fault**: For sensors and actuators, permanent faults have been considered during the FI process. After activation, permanent faults remain in the system for the rest of the execution time.

##### Fault Occurrence Rate in HVAC Systems

Two principal metrics describe the fault occurrence rate, including fault prevalence and incidence. Fault prevalence defines the fault occurrence rate of the units for a given fault at a single point in time. The fault incidence is the frequency of the fault in a specific period [59]. This paper calculates the fault occurrence rates using maintenance records and field reports based on the fault type and environmental conditions, such as the season or month the system is investigated.


**Fault Occurrence Probabilities in HVAC Systems**


Components in HVAC systems fail with different probabilities and rates due to various conditions, e.g., the number of components, environmental conditions, and unit failure rates. We have used available maintenance records to find the occurrence rates of HVAC system faults. For instance, Li et al. [41] used maintenance records to calculate the frequency and occurrence of incidents of various HVAC faults for one year. They calculated the average probability of occurrence around 0.0102 for each associated fault. Ebrahimifakhar [44] proposed the fault occurrence rates of several types of faults with different metric definitions calculated according to other FDD techniques. They also calculated average fault presence percentages for the various units, faults, and months. For instance, the average fault presence percentage of a stuck discharge air damper is estimated at approximately 8%, heating failure at 9%, and air temperature abnormality at 18% for HVAC and air handling units (AHU) in February. Faults are also listed based on their monthly presence. Hosseini Gourabpasi et al. [45] ranked HVAC-related faults and their frequencies with data-driven techniques. For example, the limit issue faults had the first rank with a rate of 15.18%. The stuck-at/partially closed faults had the second rank with a rate of 14.95%, and bias/drift/calibration faults had a probability of 10.94% and were listed in the fourth rank. Applicable unit faults in our proposed FI framework and their fault rates are listed and described in Table 4. The average probabilities for the associated fault types over one month and one day were calculated. In this paper, the fault occurrence rate during each FI is the disjoint probability of both component failure rates based on Table 4 and the application of system fault type rates. Fault type occurrence probabilities for the stuck-at fault, gain fault, offset fault, out-of-bounds and data-loss fault can be defined as 14.95%, 10.94%, 10.94%, 10.94%, 4.46%, 4.46%, respectively [44].

## 4. Implementation and Simulation 

### 4.1. Implementation

In our previous FI framework proposed in [13], a single fault was introduced out of a catalog of different fault types. In this paper, the FI framework was designed and extended to inject multiple faults in multiple zones modeled with varying faults and more dynamicity regarding the number of faults, their repetitions, and structures. Fault attributes are defined as multi-dimensional matrices such as FI time, fault duration, fault interarrival time, FI persistence, FI type, and fault occurrence probability. Each matrix element introduces the attribute values for each component and zone. By increasing the number of aspects of each attribute, the number of dimensions increases, providing multiple FI capabilities.

Figure 2 illustrates the multi-dimensional aspect of the FI framework, where the dimensions increase in the case of system model extension and development. Multi-dimensional injection attributes are explained below for the faulty rooms and components. Other attributes are activated with the same or more dimensions accordingly. For example, the “*Fault Injection Time Matrix*”elements with two fault repetitions can be assigned based on the associated faulty rooms and components defined in the “*Structure Room Component Combination Matrix*”. This matrix introduces the combinations of faulty rooms and faulty components. Figure 3 shows an example of the three-dimensional matrix for the FI attributes.



*Structure room component combination matrix (1:number of structures, 1:number of rooms, 1:number of components) = (1:1, 1:6, 1:4)*


*Fault injection time matrix (1:number of failures, 1:number of structures, 1:number of rooms, 1:number of components) = (1:2, 1:1, 1:6, 1:4)*



Figure 4 illustrates the FI timeline at the component and system levels (FCRs). Each fault can be triggered differently or at the same component at various points. A system-level timeline is a cumulative form of all FI samples in different zones.

Each fault was considered a sample event that occurs based on the probabilities of multiple faults. Each fault occurs with a specific, different, and independent probability. The fault occurrence probability was calculated based on the probabilities of the faults that happened in one unit (FCR), saved in the matrix below, and calculated according to Equation (2).
*Fault occurrence probability matrix (1:number of failures, 1:number of structures, 1:number of rooms, 1:number of components) = (1:2, 1:1, 1:6, 1:4).*
(2)$$Probability\;of\;one\;Fault\;Event=P(Failure_{1},Failure_{2},\ldots ,Failure_{i})=Unit’s\;Probability\times \left(\;{∐}_{1}^{Number\;of\;Failures}Failure_{i}\right)$$

### 4.2. Simulation

A system model can be modeled and evaluated with simulation tools. Its environmental parameters and conditions are set, and simulation results can be injected and compared with real-world scenarios [7,13]. MATLAB/Simulink, as a user-friendly tool combined with Stateflow diagrams, was used to model the behavior of the HVAC system and FI framework. Furthermore, faults were injected artificially to change the system’s behavior. An automated algorithm was coded to inject the fault attributes randomly according to the scenario-based injection type. When our algorithm runs randomly, all variables and attributes, e.g., the number of faulty components, faulty zones, and persistence, are initiated randomly.

FI blocks were designed to apply multi-dimensional matrices for activating the target faulty components and zones. Each element of the matrix in a row introduces the component indicator. If any element of one row of the matrix is 1, it can activate the faulty target zone exhibited in Figure 5.

Figure 6 shows how the faulty component is activated using a demultiplexer block to distinguish the matrix elements. For example, in Figure 6, the CO_2_ sensor and heater actuator is defective, and the damper actuator and temperature sensor are healthy. In the Stateflow diagram, with associated values and component numbers (e.g., the component number of the temperature sensor is 3), the fault is activated in the component for the next steps of the FI process.

Figure 7 shows the exterior view of the Stateflow diagram and the input parameters, including room indicator, component indicator, input data, and system model time.

The sequence of the actions and states of the FI was implemented by Stateflow diagrams using two states: the healthy state and the faulty state. Figure 8 illustrates the interior view of the Stateflow diagram shown in Figure 7 with two faulty states and one correct state. The current and active states in this figure are faulty, as indicated by the blue color. 

## 5. Evaluation and Data Analysis for Multiple Faults

In this paper, a table of scenarios was generated to evaluate the FI framework. Each fault scenario (event) was considered a sample event consisting of other sub-scenarios (sub-events). Each sub-scenario can also include multiple faults with different attribute descriptions, e.g., occurrence probabilities or fault type. To illustrate the results, two scenarios for the multiple FI and one case for more than two fault repetitions are shown in the Results Section.

### 5.1. Scenario Generation

A scenario-based approach was considered to evaluate the FI framework. To define the evaluation scenario, a Fault Injection Vector (FIV), including fault case objects, was considered and is described in Function 1. Each fault object consists of fault case attributes and faulty output data generated by the introduced automated FI algorithm. Each fault case is an object generated from *Fault_Object_Generator* class described in Table 5 comprehensively. The evaluation scenario is detailed in in Section 5.2 including the exclusive properties such as the number of FI cases, which represents the total number of injections, the number of faulty rooms, which describes the destination of the anomalies, and the number of faulty components which defines the target components in each room. Fault attributes were assigned randomly by an automatic FI algorithm. The impact shows each fault’s consequences on the system’s behavior. The effect is depicted and analyzed concerning the change ratio for each subevent and event.

**Algorithm 1** Fault Object Generator ClassClassdef Fault Object Generator  properties   *Activated_Room_component_Combination_Matrix;* //Activation of the faulty rooms and components including subevents   *Fault_Injection_Persistence_Matrix;* //Assigning the FI persistence for faulty components   *Fault_Injection_Time_Matrix;* //Assigning the FI time types   *Fault_Injection_Duration_Matrix;* //Assigning the FI duration times   *Fault_Injection_Interarrival_Matrix;* //Assigning the FI interarrival times   *Fault_Injection_Type_Matrix;* //Assigning the FI types for faulty components   *Fault_Occurrence_Probability_Matrix;* //calculating the FI types for faulty components   *Faulty_SystemOutput;* //Storing faulty output for each fault case, including system signals   *Fault_Repetitions;* //Assigning the number of repetitions for each subevent  end end

### 5.2. Evaluation and Results 

To evaluate the FI framework, the total number of 14 fault cases, including five scenarios, are defined and described in Table 6. Each scenario comprises some variations of sub-scenarios that explain the details of fault attributes and their impacts. Each fault occurrence probability value is bounded to the locality of the component, environmental conditions, and occurrence time, resulting in different CO_2_ concentrations, temperatures, and energy consumption over time. Moreover, some fault cases and their impacts are described and depicted to show the FI procedure’s accuracy and results. The impact results show the signal changes of the fault-case scenario compared with the health situation of the system model in which the up arrowhead shows an increased impact, and the down arrowhead shows a decreased impact. For some cases of intermittent faults, it was observed that the signal firstly increased and then decreased. The column of fault occurrence probability in Table 5 shows the calculated values using Equation (2). The intermittent fault cases were defined with two repetitions for the scenarios in Table 6.

#### 5.2.1. Results

Two scenarios to exhibit the thermal and energy consumption changes under faulty conditions and One to show the multiple FI for more repetitions were chosen and are illustrated below.

##### Multiple FI in Multiple Components in One Zone (Two Intermittent Stuck-at Faults in Heater Actuator and One Permanent Offset Fault in CO_2_ Sensor)

This FI case describes two component faults triggered at different points in time in one zone. One intermittent fault was activated in the heater actuator, and one permanent offset fault was initiated at the CO_2_ sensor. This scenario is related to the items 8 and 9 in Table 6.

Figure 9 shows the two stuck-at “on” faults and the stuck-at “off” faults in the heater actuator, resulting in changes in the thermal conditions. Figure 10 also depicts the CO_2_ conditions, which had a permanent offset for the rest of the execution time shown in Figure 10. Activating both faults simultaneously in one zone resulted in a reduction of temperature and a change to the “open” status of the damper actuator because with increasing CO_2_ concentration, the damper opened to decrease the harmful impact of the CO_2_. The open status of the damper actuator decreased the indoor temperature subsequently. The damper status is illustrated in Figure 11, which remained open. The open status of the damper could also cause a decrease in the CO_2_ concentration.

Figure 12 shows the energy consumption condition for this FI case which describes a substantial growth of around 73.34%. Whenever the damper stays in the open status, the heater actuator should remain on “on” to mitigate the thermal consequences and balance the internal temperature.

##### Multiple FI in Multiple Components in One Zone (Two Intermittent Stuck-at Faults in Damper Actuator and one Permanent Stuck-at Fault in Temperature Sensor)

This FI case shows two component faults in the damper actuator and temperature sensor. The damper actuator had two stuck-at “open ”and stuck-at “closed ”faults, illustrated in Figure 13. This scenario is related to the items 13 and 14 in Table 6.

The damper actuator with a stuck-at “open” fault resulted in the “on” status of the heater actuator and a reduction of the CO_2_ concentration, as shown in Figure 14 and Figure 15, respectively.

The temperature had a stuck-at value of 35 °C for the rest of the execution time, resulting in the heater actuator’s permanent “Off” status. These conditions are shown in Figure 15.

Once the heater actuator was stuck at “Off” status, this caused a remarkable reduction in energy consumption of about 67%, depicted in Figure 16. Although, as the temperature decreased, it caused thermal discomfort for the occupants.

##### Multiple Fault Injection in One Component (Intermittent Fault in Heater Actuator with 10 Repetitions)

This example scenario shows the effect of multiple faults in a single component. This intermittent fault was injected into the heater actuator with ten repetitions. Figure 17 shows the heater statuses and temperature sensor behavior. When the heater was stuck at “On,” the temperature increased, and whenever the heater was stuck at “Off,” the temperature decreased; thereby, the damper actuator subsequently became closed. The damper and heater statuses are represented in Figure 18. The number of repetitions can be dynamic and increase according to the system requirements.

Figure 19 depicts the costs due to faulty heating during the FI period. The price decreased by around 13% because the heater was stuck at closed status. It gradually decreased when the heater went to the “off” position. 

## 6. Conclusions

HVAC systems in buildings are one of the most important factors for energy consumption. Due to their vulnerabilities and complexities, they have high potential for many kinds of faults. The experimental evaluation of HVAC systems before the operational phase of the system can help designers gain insight for them to design more reliable systems. Simulation-based FI provides the opportunity to evaluate the system under various fault conditions, especially in emergency scenarios. Therefore, a fault model for multiple faults in HVAC systems based on field fault occurrence rates from maintenance records was described. Fault attributes were designed based on multi-dimensional matrices to be extensible for any system structure. A simulation-based multiple FI framework for DCV and heating systems was developed according to the defined fault model and implemented in MATLAB/Simulink using Stateflow diagrams. An automatic FI algorithm performed each fault scenario using the defined fault attributes. Different scenarios were defined to evaluate the system’s reliability and its quality indicators, such as thermal comfort, CO**_2_** concentration, energy consumption, and heating cost. Each scenario consisted of other sub-scenarios to activate multiple faults in multiple components and multiple zones. The results for the scenarios show system impacts and changes in different sub-scenarios. For example, one sub-scenario showed a rise in the heating cost and energy consumption of around 70%, and another sub-scenario exhibited a decrease in the energy consumption of around 67% but a significant increase in thermal discomfort due to the low indoor temperature. Eventually, it can be concluded that multiple FI in DCV and heating systems leads to an unexpected insight into the consequences of different fault combinations.

## 7. Future Works

This FI framework can be extended to different applications, such as stochastic FDD methods and composable system models. The FI process with statistical approaches is beneficial and applicable for FDD methods based on probabilistic and stochastic mechanisms such as Bayesian networks and pattern recognition techniques that may need initial fault probability rates and symptom frequencies to construct the network nodes and their associated paths. Our proposed framework provides beneficial data for the statistical FDD approaches. The produced output data consists of the probabilities of each fault occurrence which can be applied to any FDD technique. Furthermore, the proposed FI is a scalable and adaptable platform for any fault tolerance technique.

Future work on composable system models could focus on generating a framework that automatically extends the FI blocks and the FI structure based on an extended system model. The system model can be extended for distributed and more complex infrastructures to adapt the FI framework to these systems.

## Figures and Tables

**Figure 1 sensors-22-08180-f001:**
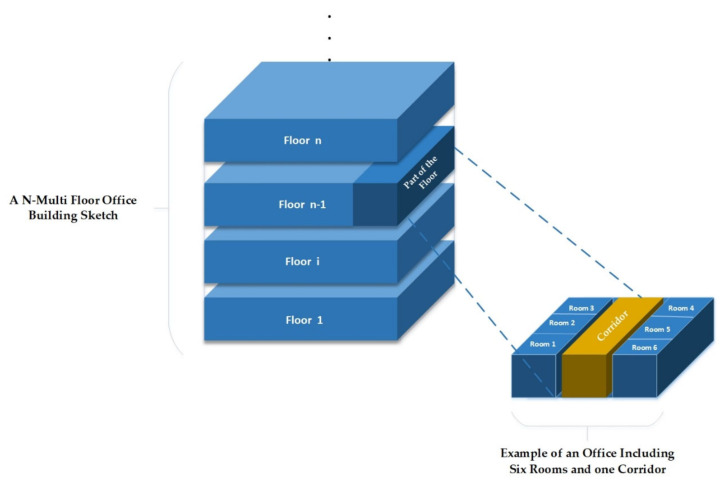
Example of a realistic office building sketch including six rooms and one corridor.

**Figure 2 sensors-22-08180-f002:**
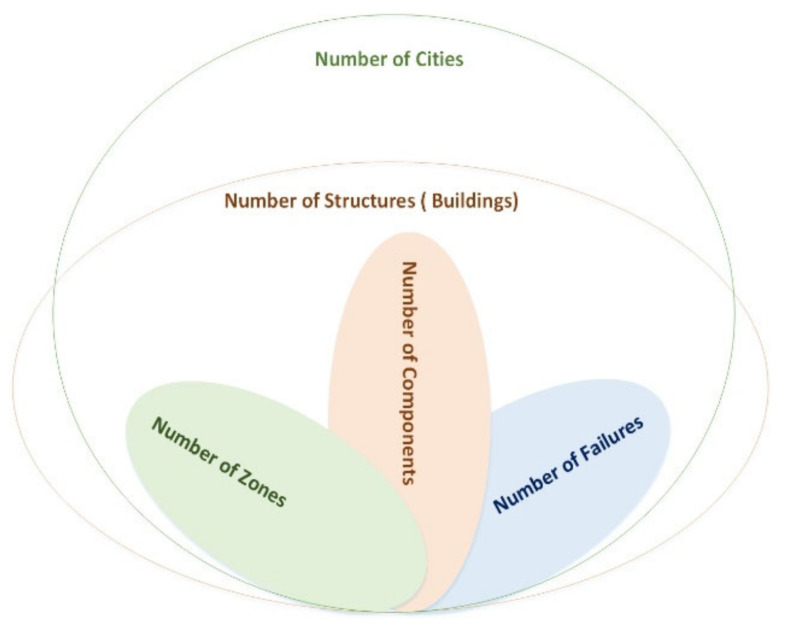
The multi-dimensional aspect of the FI implementation.

**Figure 3 sensors-22-08180-f003:**
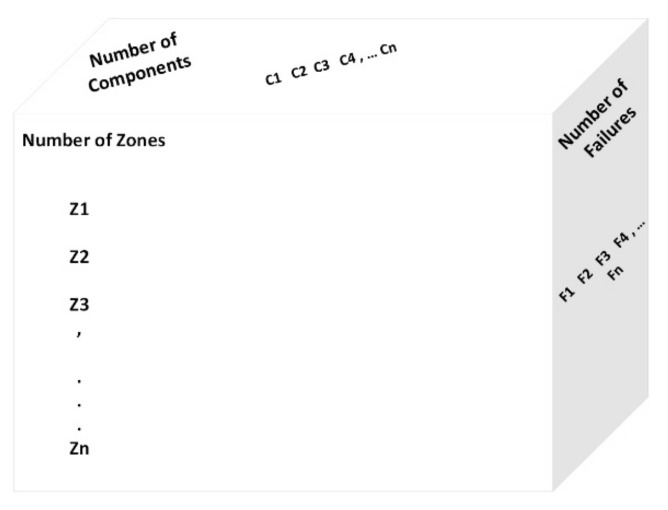
An example of the 3-dimensional matrix for the FI attributes.

**Figure 4 sensors-22-08180-f004:**
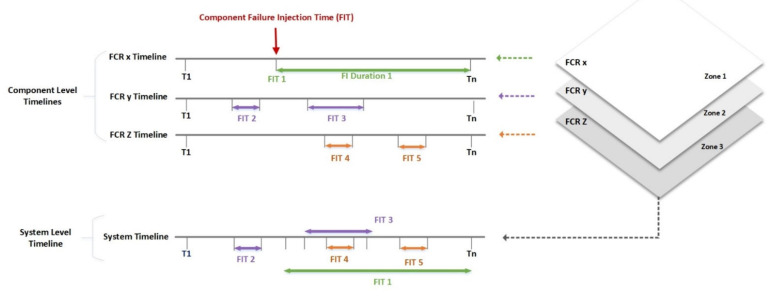
A system-level timeline for multiple-fault injection occurring in multiple FCRs in different zones.

**Figure 5 sensors-22-08180-f005:**
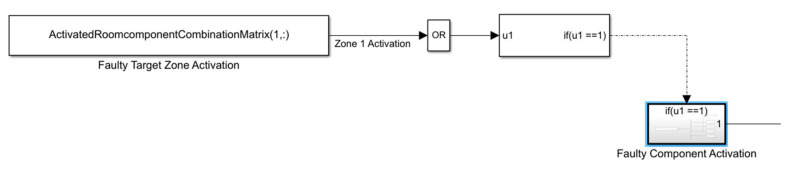
A view of faulty targeted FCR activation.

**Figure 6 sensors-22-08180-f006:**
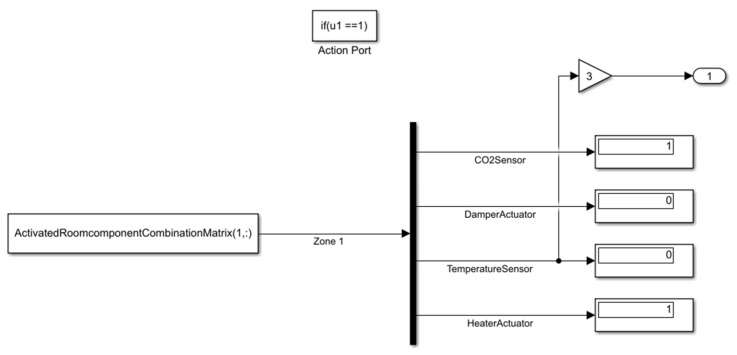
Interior view of faulty targeted component activation.

**Figure 7 sensors-22-08180-f007:**
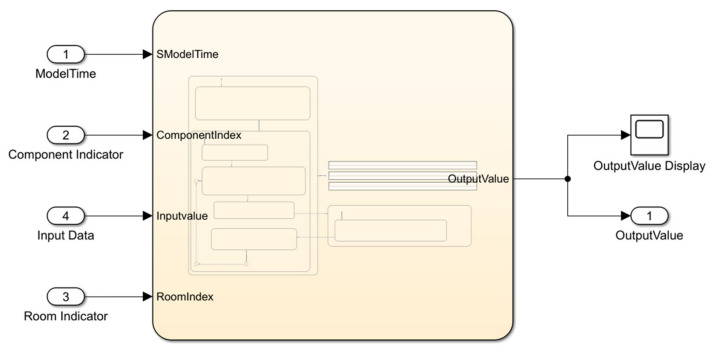
The exterior view of the Stateflow diagram.

**Figure 8 sensors-22-08180-f008:**
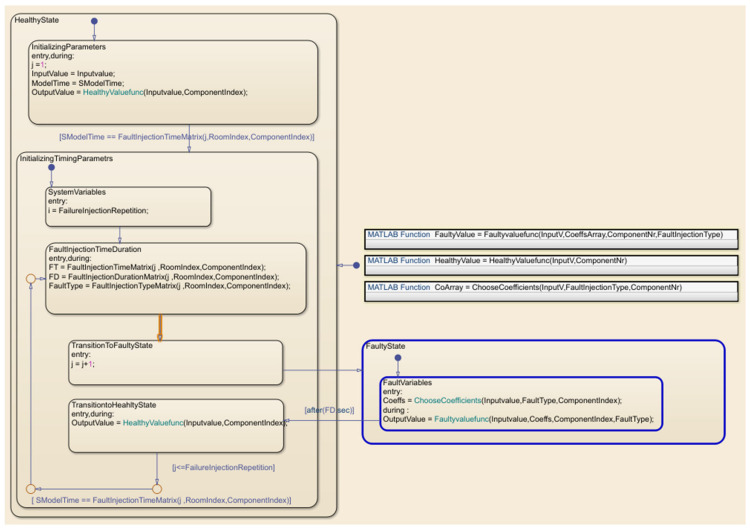
The interior view of the Stateflow diagram with two faulty and healthy states.

**Figure 9 sensors-22-08180-f009:**
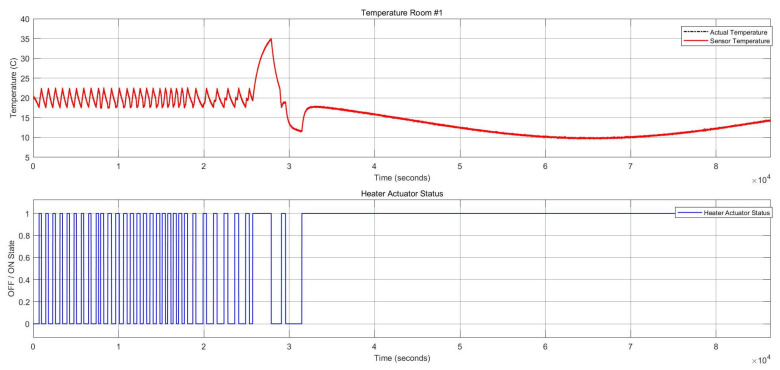
Thermal conditions for the heater actuator and CO_2_ sensor faults.

**Figure 10 sensors-22-08180-f010:**
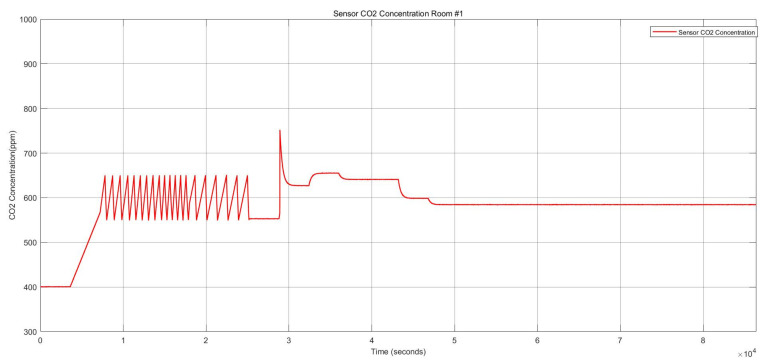
CO_2_ concentration for the heater actuator and CO_2_ sensor faults.

**Figure 11 sensors-22-08180-f011:**
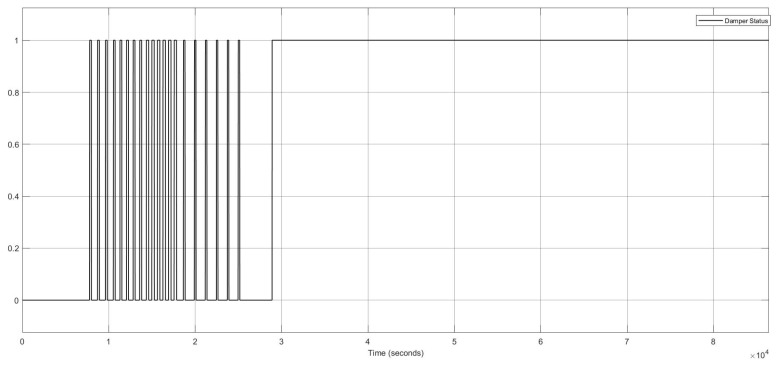
Damper actuator statuses for the heater actuator and CO_2_ sensor faults.

**Figure 12 sensors-22-08180-f012:**
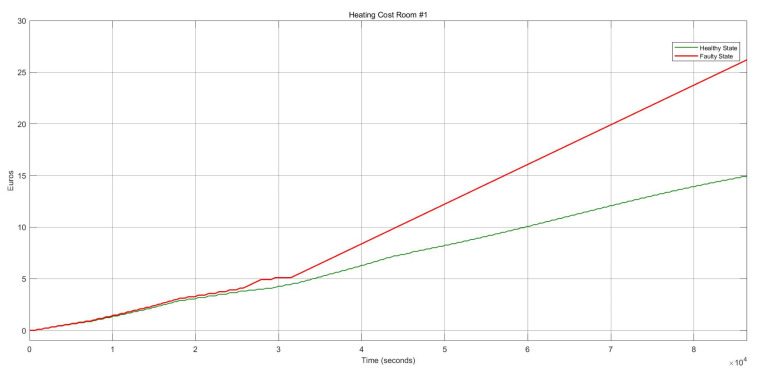
Heating cost conditions for the heater actuator and CO_2_ sensor faults.

**Figure 13 sensors-22-08180-f013:**
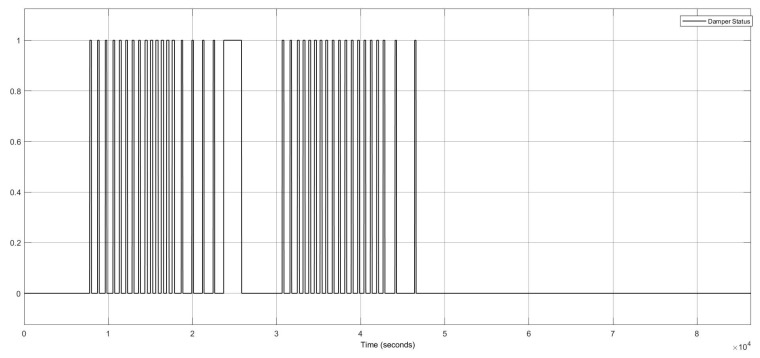
Damper actuator status for the damper actuator and temperature sensor faults.

**Figure 14 sensors-22-08180-f014:**
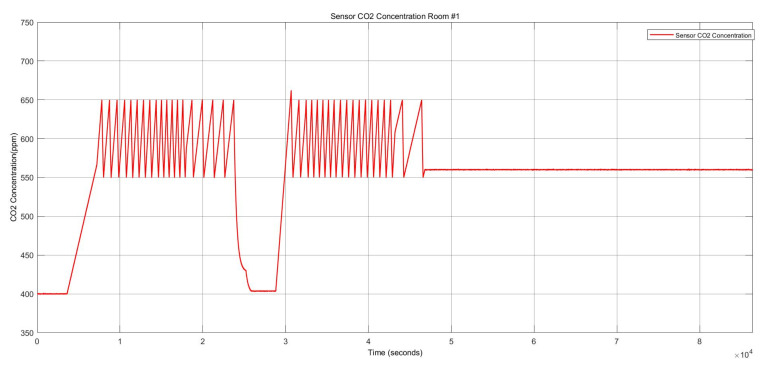
CO_2_ concentration for the damper actuator and temperature sensor faults.

**Figure 15 sensors-22-08180-f015:**
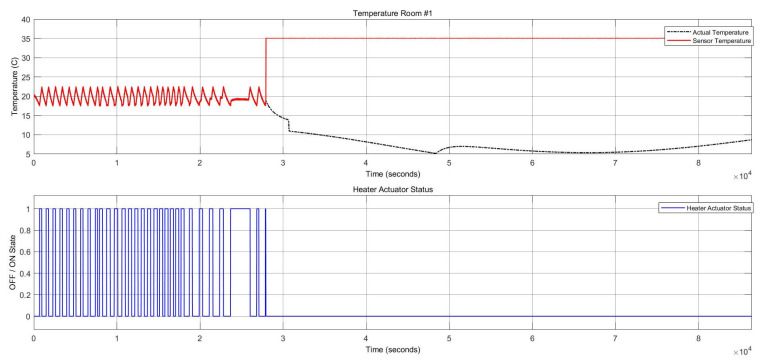
Thermal conditions and heater statuses for the damper actuator and temperature sensor faults.

**Figure 16 sensors-22-08180-f016:**
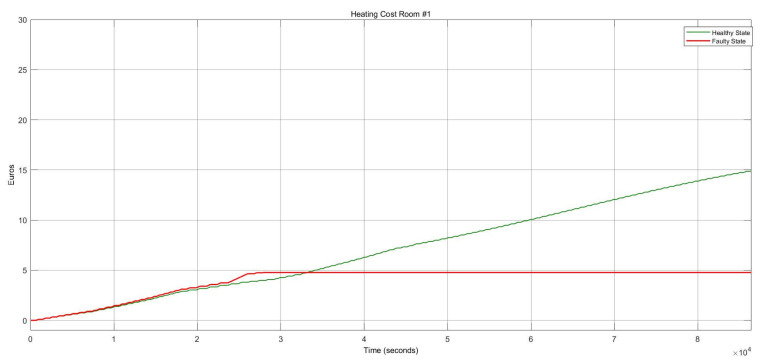
Heating cost variations for the damper actuator and temperature sensor faults.

**Figure 17 sensors-22-08180-f017:**
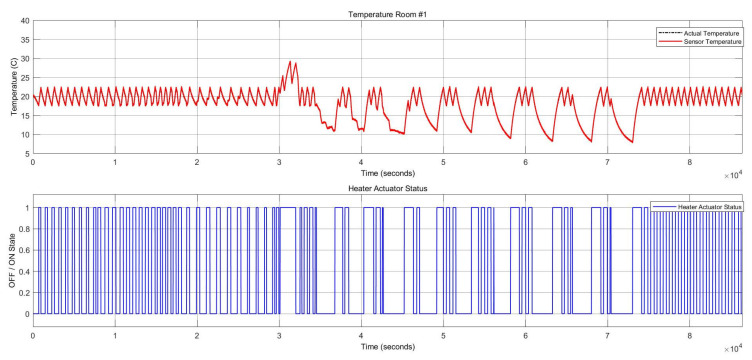
Heater actuator statuses vs. temperature sensor variations in the case of an intermittent fault with ten repetitions in the HVAC system.

**Figure 18 sensors-22-08180-f018:**
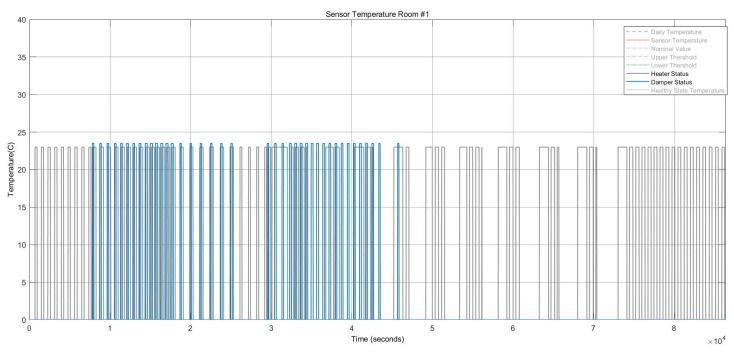
Heater statuses vs. damper statuses in the case of an intermittent fault with ten repetitions in the HVAC system.

**Figure 19 sensors-22-08180-f019:**
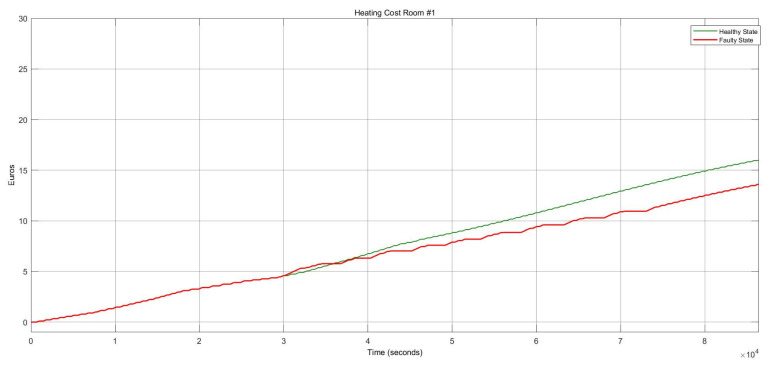
Heating cost variations for the healthy and faulty states of the HVAC system in the case of an intermittent fault with ten repetitions.

**Table 1 sensors-22-08180-t001:** The main parameters defined and used for the system modeling.

Parameters (System Variables)	Values	Units
Optimal room temperature	20	°C
Optimal CO_2_ concentration	600	ppm
Lower-threshold temperature	17.5	°C
Upper-threshold temperature	22.5	°C
Lower-threshold CO_2_ concentration	400	ppm
Upper-threshold CO_2_ concentration	700	ppm
Heater on status	1	-
Heater off status	0	-
Damper open status	1	-
Damper closed status	0	-

**Table 2 sensors-22-08180-t002:** Fault propagation examples in HVAC systems [8,55,56,57,58].

Nr.	Component Faults	Phases	Component Failure or System Fault	System Failure	Impacts
1	Wrong scheduling of the processing unit, e.g., an incorrect sequence of operations	Developmental fault: design fault	Stuck-at fault Gain fault Offset fault Out-of-bounds fault	Delay High/low/wrong sensor measurements	Equipment life Energy consumption Thermal comfort Indoor air quality
2	Programming mistakes	Developmental fault: design fault	Stuck-at fault Gain fault Offset fault Out-of-bounds fault	Delay High/low/wrong sensor measurements	Equipment life Energy consumption Thermal comfort Indoor air quality
3	Wrong setpoints too high/low	Developmental fault: design fault	Stuck-at fault Gain fault Offset fault Out-of-bounds fault	High/low/wrong temperature. High/low/wrong CO_2_ concentration	Equipment life. Occupant thermal comfort. Energy consumption
4	Oversized equipment at design phase, e.g., incorrect perimeter heating system sizing	Developmental fault: design fault	Stuck-at fault Gain fault Offset fault Out-of-bounds fault	High/low/wrong temperature. High/low/wrong CO_2_ concentration	Equipment life Occupant thermal comfort. Energy consumption
5	Improper design	Developmental fault: design fault	Stuck-at fault Gain fault Offset fault Out of bounds Data loss	Delay High/low/wrong sensor measurements	Equipment life Occupant thermal comfort. Energy consumption
6	Inaccurate location of sensors and valves, e.g., wrong thermostat location, Occupancy-sensor misplacement	Developmental fault: design fault	Stuck-at fault Gain fault Offset fault Out of bounds Data loss	Delay High/low/wrong sensor measurements	Equipment life. Occupant thermal comfort. Energy consumption
7	Missing insulation for ductwork or pipes	Developmental fault	Stuck-at fault Gain fault Offset fault	Delay High/low/wrong sensor measurements	Occupant thermal comfort. Indoor air quality
8	Poor coordination of the processing unit	Developmental fault	Stuck-at fault Gain fault Offset fault Out of bounds Data loss	Delay Missing information	Occupant thermal comfort. Indoor air quality Delay
9	Air-duct leakages	Operational faults	Stuck-at fault Gain fault Offset fault	Wrong actuator signals	Equipment life Thermal discomfort Indoor air quality Energy consumption
10	Inappropriate voltage	Operational faults	Stuck-at fault Gain fault Offset fault Out of bounds Data loss	Wrong actuator signals High/low/wrong sensor measurements Missing information	Equipment life Thermal discomfort Energy Consumption Life risk Fire risk
11	Poor preventive maintenance	Operational faults	Stuck-at fault Gain fault Offset fault Out of bounds Data loss	Delay Missing information	Equipment life Energy consumption Life risk Fire risk

**Table 4 sensors-22-08180-t004:** The faults and their fault occurrence incidents for the associated fault types.

Nr.	Component	System Faults	Average Presence of Faults in February	Average Monthly Presence of Faults Among the Total of 28 Faults	Total Monthly Probability	Total Daily Probability
1	**Temperature Faults**	Temperature sensor fault	18%	8%	0.2538	0.0091
		Temperature frozen		35%		
		The mismatch between supply air temperature and its setpoint		26%		
		Supply air temperature abnormal		12%		
		Mix air temperature sensor fault		4%		
		Mix air temperature abnormal		22%		
		Return air temperature abnormal		2%		
		Setpoint fault		4%		
		Missed control optimization		28%		
2	**Heater Faults**	Heater abnormality	9%	2%	0.0324	0.001157
		Heating coil valve leakage		2%		
		Setpoint fault		4%		
		Missed control optimization		28%		
3	**CO_2_ Faults**	Airflow sensors abnormalities (CO_2_ sensor)	13%	10%	0.05785	0.00206
		Return airflow abnormal		1.5%		
		Return air CO_2_ sensor		1%		
		Missed control optimization		28%		
		Setpoint fault		4%		
4	**Damper Faults**	Damper stuck	8%	11%	0.0312	0.0028
		Missed control optimization		28%		

**Table 5 sensors-22-08180-t005:** FIV consists of fault-case objects.

Fault_Case Obj1	Fault_Case Obj2	Fault_Case Obj	..	Fault_Case Obji	..	Fault_Case Obj n-1	Fault_Case Objn

**Table 6 sensors-22-08180-t006:** Scenario descriptions for the FI framework in the HVAC system.

Nr.	Scenarios (Events)	Sub Scenarios (Sub Event)	Faulty Room	Faulty Components	Faults’ Attributes				Impacts		
					Fault Persistence	First Fault Type	Second Fault Type	Fault Occurrence Probability	CO_2_ Concentration Changes (ppm)	Temperature Changes (°C)	Energy Changes
1	1	1	1	Damper actuator	Intermittent	Stuck-at (open)	Stuck-at (closed)	0.6258	↑	−	↑
2		2	4	CO_2_ sensor	Permanent	Gain fault	−	0.0225	↑	↓	↑
3	2	1	2	Temperature sensor	Permanent	Out of bounds	−	0.0996	−	↓	↓
4		2	5	Heater actuator	Intermittent	Stuck-at (on)	Stuck-at (off)	0.2586	−	↑↓	↑
5	3	1	4	Damper actuator	Intermittent	Stuck-at (open)	Stuck-at (closed)	0.6258	↓↑	−	↑
**6**		2	5	Temperature sensor	Permanent	Out Of bounds	−	0.0996	−	↑	↓
**7**		3	5	Heater actuator	Intermittent	Stuck-at (off)	Stuck-at (on)	0.2586			
8	4	1	4	CO_2_ sensor	Permanent	Offset fault	−	0.0225	↑	↑↓	↑
9		2	4	Heater actuator	Intermittent	Stuck-at (on)	Stuck-at (off)	0.25856			
10		3	5	CO_2_ sensor	Permanent	Offset fault	−	0.0225	↑	↓	↑
11		4	5	Damper actuator	Intermittent	Stuck-at (open)	Stuck-at (open)	0.6258			
12	5	1	1	CO_2_ sensor	Permanent	Stuck-at	−	0.0308	↓	−	↓
13		2	5	Temperature sensor	Permanent	Out of bounds	−	0.6258	↓	↑	↓
14		3	5	Damper actuator	Intermittent	Stuck-at (open)	Stuck-at (closed)	0.0996			

## Data Availability

Not applicable.

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
