# Peer review of "Fault Injection with Multiple Fault Patterns for Experimental Evaluation of Demand-Controlled Ventilation and Heating Systems"

_sensors, 2022, doi:10.3390/s22218180_

Round 1

Reviewer 1 Report

The paper presents a fault injection framework for injecting multiple sensors and actuator faults with realistic fault modes and failure rates from field reports. Specifically, a multi-dimensional fault model is defined by considering the probability of occurrence of different types of sensor and actuator faults. This could be practically useful for development testing.  But the paper does not have sufficient material in terms of novelty and technical quality.

Some specific comments:

There are many wording and phrasing issues, which easily cause confusion.

Contributions on page 3 are not clear and should be stated with respect to the existing work.

For Figure 1, the legend said: “Occupany” Sensor.

In Table 2, the units do not align with parameters. ppm is used for temperature while Celsius is used for CO2 concentration.

Eq.1 does not have any notations.

Author Response

Dear Sir,

We thank the reviewers and the editor-in-chief for the valuable feedback.

In the following, we would like to point out how the comments have been incorporated into the updated version of the manuscript. Please find our responses in blue color.

Best regards,

Ali Behravan

Bahareh Kiamanesh

Roman Obermaisser

  • Comments from Reviewer

     There are many wording and phrasing issues, which easily confuse.

The manuscript was edited by the authors.

  1. Contributions on page 3 are not clear and should be stated with respect to the existing work.

Contributions are updated as described below.

This paper provides contributions toward a dependability evaluation of HVAC systems faced with multiple faults at the same time:

  • Modeling patterns of multiple faults in HVAC systems based on data from field failure rates and maintenance records: The paper maps insights from maintenance records to FI patterns with multiple faults. Fault occurrence probability is an important parameter to design a realistic FI framework because this parameter is affected by environmental conditions, e.g., dust and dirt, seasons and respective temperatures, working conditions, application areas, and the locality of faults in various components of a system. Therefore, a fault model is created using statistical parameters such as fault occurrence probability. Fault occurrence probabilities enable defining the scenario cases and performing FI based on different environmental conditions and fault type rates.
  • Injecting multiple faults into an HVAC system according to the fault models: The paper introduces a FI framework with FI vectors that provide exact control of the timing, locality, and values of multiple faults. The FI framework is designed to inject multiple faults in multiple zones and multiple components with corresponding faults’ attributes. Faults are initiated by an automatic FI algorithm. Faults’ repetitions and multi-dimensional fault attributes can be assigned associatively randomly. The framework is generic, and the vectors can be customized to different buildings. The paper shows how fault patterns for multiple faults can be established for a particular building based on maintenance records from the related work.
  • Experimentally evaluating the effects of multiple faults on the behavior of the HVAC systems: The paper provides comprehensive experimental results and insights into the system behavior upon multiple faults using the patterns of multiple faults. Due to the use of real-world statistics and maintenance records, the results are realistic. This is a significant result of research on fault management techniques coping with multiple faults, for which no experimental data is available today.
  1. For Figure 1, the legend said: “Occupancy” Sensor.

Figure 1 It illustrates a N-Multi floor office building. The part of one floor, that is our simulated target model has been shown as an example of an office with six rooms and one corridor.

  1. In Table 2, the units do not align with the parameters. ppm is used for temperature while Celsius is used for CO2 concentration.

Table 2 is changed to Table 1 and edited.

  1. 1 does not have any notations.

Notations for equation 1 are added and explained.

Author Response

Dear Sir,

We thank the reviewers and the editor-in-chief for the valuable feedback.

In the following, we would like to point out how the comments have been incorporated into the updated version of the manuscript. Please find our responses in blue color.

Best regards,

Ali Behravan

Bahareh Kiamanesh

Roman Obermaisser

  • Comments from Reviewe
  1. It is difficult to find anything new in this manuscript. What is the main difference between this and the given below article?

This paper provides contributions toward a dependability evaluation of HVAC systems faced with multiple faults at the same time:

  • Modeling patterns of multiple faults in HVAC systems based on data from field failure rates and maintenance records: The paper maps insights from maintenance records to FI patterns with multiple faults. Fault occurrence probability is an important parameter in designing a realistic FI framework because this parameter is affected by environmental conditions, e.g., dust and dirt, seasons and respective temperatures, working conditions, application areas, and the locality of faults in various components of a system. Therefore, a fault model is created using statistical parameters such as fault occurrence probability. Fault occurrence probabilities enable defining the scenario cases and performing FI based on different environmental conditions and fault type rates.
  • Injecting multiple faults into an HVAC system according to the fault models: The paper introduces a FI framework with FI vectors that provide exact control of the timing, locality, and values of multiple faults. The FI framework is designed to inject multiple faults in multiple zones and multiple components with corresponding faults’ attributes. Faults are initiated by an automatic FI algorithm. Faults’ repetitions and multi-dimensional fault attributes can be assigned associatively randomly. The framework is generic, and the vectors can be customized to different buildings. The paper shows how fault patterns for multiple faults can be established for a particular building based on maintenance records from the related work.
  • Experimentally evaluating the effects of multiple faults on the behavior of the HVAC systems: The paper provides comprehensive experimental results and insights into the system behavior upon multiple faults using the patterns of multiple faults. Due to the use of real-world statistics and maintenance records, the results are realistic. This is a significant result of research on fault management techniques coping with multiple faults, for which no experimental data is available today.

2. I did not find any new steps in this manuscript. For example, the fault model, which is given in equation (1) is the same as above in the mentioned article.

An extension of the presented fault model is the fault occurrence rates which are calculated using maintenance records.

  1. This manuscript lacks novelties. In Table 1 (On page 7, line 214), an extension of the fault injection is not sufficient.

The paper has been updated and to add some paragraphs to highlight the contribution from line 59 to 85 (page two).

4. The presented articles in Table 1 (On page 2, lines 71-72), are not too updated. The authors investigated the fault type, Fault persistence, Fault Interarrival, etc. From this analysis, it seems the investigated problem is not the latest.

Table 1 was deleted, and the related work section was extended to present more recent references.

  1. Many figures exist in the literature, for example, Figures 1, 12, 16, and 19, etc. I did not find any newly developed analysis in the simulation section. The authors used the same scenario from their previous publication in Figure 1. However, Figures 12, 16, and 19 explore the heating cost conditions for the heater actuator and sensor faults which would not be acceptable.

 Figure 1 is changed.

  • Section 5.2.1.2 their figures were changed because section 5.2.1.2 shows multiple fault injections for two components.

 Figure 13, 14, 15, and 16 have changed and show multiple fault injection in multiple components in one zone (two intermittent stuck at fault in damper actuator and one permanent stuck at fault in temperature sensor)

  • Section 5.2.1.3 their figures were changed because of an error for the last failure injection.

 Figures 17, 18, and 18 have changed and represent multiple fault injections in one component (intermittent fault in heater actuator with ten repetitions) 

  • Scenario generation is based on multiple fault injections, including events and subevents, which differ from previous papers that focused on single fault injection.

In this paper, a table of scenarios has been generated to evaluate the FI framework. Each fault scenario (event) has been considered a sample event consisting of other sub-scenarios (sub-events). Each sub-scenarios can also include multiple faults with different attribute descriptions, e.g., occurrence probabilities or fault type

Reviewer 3 Report

Review of the manuscript

Manuscript No: Sensors-1899296

Title: Multiple-Fault Injection for Evaluation of Demand-Controlled Ventilation and Heating Systems

Authors: Bahareh Kiamanesh, Ali Behravan and Roman Obermaisser

Review report

In this manuscript, authors reported the Multiple-Fault Injection for Evaluation of Demand-Controlled Ventilation and Heating Systems. The aim of the work meets requirement of Sensors, but there will be some questionable points in the manuscript. The manuscript needs some improvement in the language and also in the grammatical construction of the sentences as the language does not meet the standards of the Journal (Sensors). The research and technical explanations are good. Hence minor revision is required. The manuscript can be revised based on the following review comments.

Review comments.

1.      Authors have done the good work. This manuscript needs some improvement in the language and also in the grammatical construction of the sentences. Authors must check the language quality and grammatical/typo errors in throughout the manuscript.

2.      Authors have performed the numerical/simulation work which must be addressed in the title, abstract and conclusions.

3.      Actual temperature data is not visible in Figure 9. If it is not required, remove it from the legend of the graph. Follow the same for remaining graphs.

4.      Quantitative results can be added in the abstract and conclusion.

5.      Quality of the figures can be improved.

6.      Mechanisms can be improved in the discussion section.

7.      Try to avoid more self citations. Few self-citations can be used.

8.      The study was good and paper written well. I recommend this paper for publication with minor revision

Author Response

Dear Sir,

We thank the reviewers and the editor-in-chief for the valuable feedback.

In the following, we would like to point out how the comments have been incorporated into the updated version of the manuscript. Please find our responses in blue color.

Best regards,

Ali Behravan

Bahareh Kiamanesh

Roman Obermaisser

  • Comments from Reviewer
  1. The authors have done good work. This manuscript needs some improvement in the language and also in the grammatical construction of the sentences. Authors must check the language quality and grammatical/typo errors throughout the manuscript.

The language in the manuscript was improved by the authors, as suggested.

  1. Authors have performed the numerical/simulation work which must be addressed in the title, abstract, and conclusions.

Numerical and simulation results have been added in the abstraction and conclusion parts

The title is also changed to “Fault Injection with Multiple Fault Patterns for Experimental Evaluation of Reliability and System Quality in Demand-Controlled Ventilation and Heating Systems.”

  1. Actual temperature data is not visible in Figure 9. If it is not required, remove it from the legend of the graph. Follow the same for the remaining graphs.

The actual temperature in figure 9 is active as this is bold. So it means that it shows that the actual temperature and measured temperature sensor are the same in the 5.2.1.1 example scenario.

  1. Quantitative results can be added to the abstract and conclusion.

Quantitative results have been added in the abstraction and conclusion parts.

  1. The quality of the figures can be improved.

All figures have been replaced with figures with more quality.

  1. Mechanisms can be improved in the discussion section.

The conclusion and discussion part has been updated by adding more details about the mechanism.

HVAC systems in buildings are one of the most important factors for energy consumption and, due to their vulnerabilities and complexities, have a high potential for many kinds of faults. The experimental evaluation of HVAC systems before the operational phase of the system can help designers gain insights and to design more reliable systems. Simulation-based FI provides the opportunity to evaluate the system under various fault conditions, especially in emergency scenarios. Therefore, a fault model for multiple faults in HVAC systems based on field fault occurrence rates from maintenance records has been described. Fault attributes have been designed based on multi-dimensional matrices to be extensible for any system structure. A simulation-based multiple fault injection framework for DCV and heating systems was developed according to the defined fault model and implemented in MATLAB/Simulink using Stateflow diagrams. An automatic fault injection algorithm performs each fault scenario using the defined fault attributes. Different scenarios were defined to evaluate the system reliability and system quality indicators such as thermal comfort, CO2 concentration, energy consumption, and heating cost. Each scenario consists of other sub-scenarios to activate multiple faults in multiple components and multiple zones. The results for the scenarios show system impacts and changes in different sub-scenarios. For example, one sub-scenario shows a rise in the heating cost and energy consumption of around 70%, and another sub-scenario exhibits a decrease in the energy consumption of around 67% but a significant increase in thermal discomfort due to the low indoor temperature. Eventually, it can be concluded that multiple fault injection in DCV and heating systems leads to unexpected insights into the consequences of different fault combinations.

  1. Try to avoid more self-citations. Few self-citations can be used.

The related work section is added,  and more recent and relevant references were included.

  1. The study was good, and the paper was written well. I recommend this paper for publication with minor revision

Round 2

Reviewer 1 Report

The revised version reads much better.  But the claimed contributions of the paper are limited. Overall, the paper lacks novelty.

Author Response

Dear Reviewer,

Thank you very much for your valuable feedback in the 2nd round of the review process. We extended the related work section by highlighting the research gaps and the contributions of the paper beyond the state of the art in the following three areas:

(1) Modeling Patterns of Multiple Faults in DCV and Heating Systems Based on Data From Field Failure Rates and Maintenance Records
(2) Injecting multiple faults into a DCV and heating system according to the fault models
(3) Experimentally Evaluating the Effects of Multiple Faults on the Behavior of DCV and Heating Systems

Please refer to the highlighted sections in yellow color in paper sections 2.5.1, 2.5.2, and 2.5.3.  

Best regards,

Bahareh Kiamanesh, Ali Behravan, Roman Obermaisser

Reviewer 2 Report

Now the revision of this manuscript is fine.

Author Response

(The authors gave the same response as above.)
